# Perspectives on microRNAs and Phased Small Interfering RNAs in Maize (*Zea mays* L.): Functions and Big Impact on Agronomic Traits Enhancement

**DOI:** 10.3390/plants8060170

**Published:** 2019-06-12

**Authors:** Zhanhui Zhang, Sachin Teotia, Jihua Tang, Guiliang Tang

**Affiliations:** 1State Key Laboratory of Wheat and Maize Crop Science, Henan Agricultural University, Zhengzhou 450002, China; steotia@mtu.edu (S.T.); tangjihua1@163.com (J.T.); 2Department of Biological Sciences, Michigan Technological University, Houghton, MI 49931, USA; 3Department of Biotechnology, Sharda University, Greater Noida 201306, India

**Keywords:** maize (*Zea mays* L.), miRNA, phasiRNA, tasiRNA, agronomic traits, crop improvement

## Abstract

Small RNA (sRNA) population in plants comprises of primarily micro RNAs (miRNAs) and small interfering RNAs (siRNAs). MiRNAs play important roles in plant growth and development. The miRNA-derived secondary siRNAs are usually known as phased siRNAs, including phasiRNAs and tasiRNAs. The miRNA and phased siRNA biogenesis mechanisms are highly conserved in plants. However, their functional conservation and diversification may differ in maize. In the past two decades, lots of miRNAs and phased siRNAs have been functionally identified for curbing important maize agronomic traits, such as those related to developmental timing, plant architecture, sex determination, reproductive development, leaf morphogenesis, root development and nutrition, kernel development and tolerance to abiotic stresses. In contrast to *Arabidopsis* and rice, studies on maize miRNA and phased siRNA biogenesis and functions are limited, which restricts the small RNA-based fundamental and applied studies in maize. This review updates the current status of maize miRNA and phased siRNA mechanisms and provides a survey of our knowledge on miRNA and phased siRNA functions in controlling agronomic traits. Furthermore, improvement of those traits through manipulating the expression of sRNAs or their targets is discussed.

## 1. Introduction

Plant and animal small RNAs (sRNAs) are short noncoding regulatory RNAs in the size range of ~20 to 30 nucleotides (nt) [1,2]. These sRNAs play crucial roles in various biological regulatory processes through mediating gene silencing at both transcriptional and posttranscriptional levels [1,3]. According to the origin and biogenesis, plant sRNAs can be categorized into several major classes, micro RNAs (miRNAs), heterochromatic small interfering RNAs (hc-siRNAs), phased small interfering RNAs (phased siRNAs), and natural antisense transcript small interfering RNAs (NAT-siRNAs) [4].

Plants miRNAs are processed from long *MIRNA* transcripts by a microprocessor and dicing complexes [5,6,7]. Compared to animal miRNAs, plant miRNAs tend to have fewer targets that mainly encode transcription factors and F-box proteins [8]. This indicates that miRNA is at the central position of gene expression regulatory networks of plant growth and development. The accumulating studies proved miRNAs to be key regulators of various biological regulatory processes in plants, including developmental timing, plant architecture, organ polarity, inflorescence development and responses to biotic and abiotic stresses [9,10]. Additionally, miRNAs also drive secondary siRNA generation that are defined as phased siRNAs. Such secondary siRNAs, including canonical phased siRNAs (phasiRNAs) and phased trans-acting siRNAs (tasiRNAs), also play key roles in plant development [11,12,13]. Moreover, manipulation of mRNA transcript abundance via miRNA control provides a unique strategy for the improvement of the complex agronomic traits of crops [14,15]. Thus, understanding the functions of miRNAs and related secondary siRNAs in various plant species, especially in crops like maize, is essential for crop improvement.

Maize is not only a model plant genetic system, but is also an important crop species for food, fuel and feed [16]. Like *Drosophila* and the worm *Caenorhabditis elegans*, maize has been a significant contributor to a number of important discoveries, including the so-called “jumping genes” (transposons), activator/dissociation (Ac/Ds) and Mutator, as well as the epigenetic phenomenon termed paramutation [17,18]. However, there have been limited studies on the roles of miRNAs and miRNA-derived secondary siRNAs in maize metabolism, development and stress responses [12,19,20,21,22,23,24,25,26,27,28,29,30,31,32,33,34], making it far from utilized in agronomic traits improvement through genetic engineering. Compared to *Arabidopsis* and rice, maize sRNA and RNAi mechanisms remain only partially resolved.

This review examines the current status of our understanding of the biogenesis and functions of miRNA and phased siRNA in maize, with a focus on their key components and the missing links of the pathways. Such study can help evaluate the potential roles of maize sRNAs in the enhancement of agronomic traits. First, we survey the recent findings regarding miRNA and phased siRNA working mechanisms in maize. We further compare the mechanistic differences for those mechanisms between maize and the model plants *Arabidopsis* and rice highlighting the missing links in maize. Furthermore, we review the identified miRNA and phased siRNA functions in regulating important agronomic traits in maize. Finally, we discuss the potential applications of these small regulatory RNAs or of their target genes in agronomic traits enhancement.

## 2. MiRNA and phasiRNA Biogenesis in Maize

### 2.1. Core Components of sRNA Biogenesis in Plants

In plants, the sRNAs biogenesis and gene silencing mainly depends on the activities of three kinds of proteins, dicer or dicer-like proteins (DCLs), argonautes (AGOs) and RNA-dependent RNA polymerases (RDRs) [1,32,35,36]. The sRNA-mediated gene silencing is initiated by double stranded RNA (dsRNA) generation by RDRs or the folding of *MIRNA* gene transcripts [1,36]. The dsRNA is processed into sRNAs, 20–30 nt in length, by the cleavages of microprocessors, DCLs [1,5,36]. Different classes of sRNAs are recognized by specific AGOs to assemble RNA-induced silencing complex (RISC) [1,36]. In *Arabidopsis*, 4 DCLs, 10 AGOs and six RDRs are encoded, while in rice, eight DCLs, 19 AGOs and five RDRs are encoded [35]. In maize, these major components have also been identified mainly based on their orthologs in *Arabidopsis* and rice genomes, which include five ZmDCL, 17 ZmAGO and five ZmRDR genes (Table 1, Figure 1A) [32,37]. Among them, only a few ones have been experimentally verified so far, including *fuzzy tassel* (*fzt, ZmDCL1*) [22], *ragged seedling2* (*rgd2, ZmAGO7*) [38], *ZmRDR1* [37], *mediator of paramutation 1* (*mop1*, *ZmRDR2*) [39].

### 2.2. MiRNA-Mediated Gene Silencing in Maize

In plants, 21-nt miRNA biogenesis includes four steps (Figure 1): (1) *MIRNA*s transcription; (2) precursor miRNA (pre-miRNA) generation by dicer-like RNase III protein I (DCL1) cleavage; (3) miRNA duplexes release; (4) miRNA duplexes methylation and export to cytoplasm and miRNA-RISC assembly [6,7,40,41,42]. In the cytoplasm, the miRNA-RISCs mediate their target mRNA degradation [43,44], or translational inhibition in plants [45]. In contrast to the 21 nt miRNAs, a class of 24 nt miRNAs was discovered in plants (Figure 1). These 24 nt miRNAs are processed by DCL3 during their biogenesis [46,47]. In RNA-directed DNA methylation (RdDM), these 24 nt miRNAs are sorted into AGO4 to direct DNA methylation at the loci of their origin, thus regulating their target genes in trans [46,47]. After their biogenesis, miRNAs are also subjected to catabolism [48], in which demethylated or uridylated miRNAs are degraded by small RNA degrading nucleases (SDNs) [40,49].

### 2.3. Origin and Biogenesis of phasiRNAs in Maize

Generally, phasiRNA biogenesis is initiated by cleavage of single-stranded *PHAS* loci transcripts by 22 nt miRNAs. Then, those cleaved single-stranded RNAs are used to generate dsRNAs by RDRs. DCLs further phase dsRNAs to produce 21 or 24 nt phasiRNAs. PhasiRNAs are subsequently loaded to AGOs to regulate gene expression network [11,50] (Figure 2A,B). In grasses, including maize, phasiRNA precursors, *PHAS* loci transcripts, are transcribed by RNA polymerase II. These long noncoding precursor transcripts are internally cleaved, guided by 22 nt miR2118 to generate the 21 nt phasiRNAs or by miR2275 for the 24 nt phasiRNA. Such special class of small RNAs are specifically expressed in reproductive organs, conferring male fertility [11,12,51]. In maize, the biogenesis of 21 and 24 nt phasiRNAs are regulated by DCL4 and DCL5, respectively (Figure 2A,B). Next, 21 and 24 nt phasiRNAs are recruited by AGO5c and AGO18b, respectively, to assemble RISC and regulate gene expression [12].

The production of the 21-nt tasiRNAs is initiated by a miRNA through RDR6 and DCL4 [52]. In *Arabidopsis*, miR173, miR390 and miR828 trigger the production of *TAS1a-c*/*TAS2*, *TAS3*, and *TAS4* siRNAs, respectively [53,54]. The maize *TAS3* pathway has been identified through the mutations, *leafbladeless1* (*ldl1*) and *ragged seedling2* (*rgd2*), which encode the orthologs of SGS3 and AGO7 of *Arabidopsis* (Figure 2C) [55]. After *TAS3* siRNAs is generated, they are recruited by AGO7 to assemble RISC and induce ARF3 gene silencing by targeting mRNA transcripts.

### 2.4. Functional Redundancy and Divergence of the Key Components in Maize sRNA Biogenesis Pathways

#### 2.4.1. DCLs

Based on the phylogenetic analysis, different DCLs from *Arabidopsis*, rice and maize were classified into four subgroups (Table 1, Figure 3A,B). ZmDCL1 showed high similarity with *Arabidopsis* AtDCL1 and rice OsDCL1a–1c; ZmDCL3a and ZmDCL5/3b are similar to *Arabidopsis* DCL3 and rice OsDCL3a–3b; and ZmDCL2 and ZmDCL4 are most similar to AtDCL2 and AtDCL4, respectively [23,32]. In *Arabidopsis*, AtDCL1 produces mature miRNAs [56]; AtDCL2 is involved in virus defense-related siRNA generation and has functional redundancy with AtDCL4 [57]; while AtDCL3 catalyzes the production of 24-nt siRNAs [58]; and AtDCL4 is mainly for the production of tasiRNAs [59]. Although the DCL family proteins are largely functionally conserved among the three plant species, DCL3a and DCL3b are considered specific to monocots and predate the divergence of rice and maize [60].

#### 2.4.2. AGOs

In *Arabidopsis*, AtAGO1 is associated with miRNA-mediated gene silencing [61]; AtAGO7 is preferentially associated with a single miRNA, miR390, to trigger production of *TAS3* [52]; and AtAGO5 is a putative germline-specific Argonaute complex associated with miRNAs in mature *Arabidopsis* pollen [62]. In addition, AtAGO2 was identified to have a stand-in role for AtAGO1 in antivirus defense when AGO1-targeted silencing is overcome by viral suppressors [63], AtAGO4 is associated with endogenous siRNAs that direct DNA methylation [64].

In maize, 17 genes encoding 18 AGO family proteins were identified, almost double the number reported in *Arabidopsis* (Table 1, Figure 3A,C) [24,32]. These ZmAGOs were divided phylogenetically into five subgroups: AGO1 (ZmAGO1a-1d and ZmAGO10a, b), MEL1/AGO5 (ZmAGO5a-5d), AGO7 (ZmAGO2 and ZmAGO7), AGO4 (ZmAGO4), and finally the ZmAGO18 (ZmAGO18a-c) [32]. The ZmAGO18 subgroup, ZmAGO18a, ZmAGO18b and ZmAGO18c, are encoded by two genes (*GRMZM2G105250* encodes ZmAGO18a, and ZmAGO18b and ZmAGO18c are encoded by two transcripts of *GRMZM2G457370*) [32]. They displayed high structural similarity to OsAGO18, whose expression is strongly induced by viral infection in rice and confers broad-spectrum virus resistance by sequestering the OsmiR168 from targeting OsAGO1 [65]. Nonetheless, ZmAGO18a is highly expressed in ears, while ZmAGO18b is mostly enriched in tassels, suggesting that ZmAGO18 family may have functional diversities from the OsAGO18 [66]. In fact, ZmAGO18b was proposed to bind the 24 nt phasiRNAs that are suggested to be the products of ZmDCL5/3b in the phasiRNA pathway, based on their concurrent spatial and temporal expression in developing maize ear/tassel development [12]. The mutant *ragged seedling2* (*rgd2*) has been identified to encode an AGO7-like protein required to produce *TAS3* [38], and its functions are highly conserved among *Arabidopsis*, rice and maize [67,68,69].

#### 2.4.3. RDRs

Six, five and five RDRs have been identified in *Arabidopsis*, rice and maize, respectively (Table 1, Figure 1D) [32,35]. These RDRs were divided phylogenetically into four subgroups: RDR1, RDR2, RDR3/4/5, and RDR6. AtRDR1 and its homolog in maize, ZmRDR1, have been reported to be involved in antiviral defense [37]. AtRDR2 plays a crucial role in RNA-directed DNA methylation and repressive chromatin modifications of certain transgenes, endogenous genes and centromeric repeats that correlate with the production of 24 nt interfering sRNAs [72]. In maize, MOP1 (a homolog of AtRDR2) has proven to be essential for a siRNA-directed gene-silencing pathway, and is also involved in the maintenance of transposon silencing and paramutation [39]. The remaining three RDR homologs of *Arabidopsis* RDR6 in maize, ZmRDR6a-c, are involved in tasiRNA biogenesis [67,73]. We tentatively renamed these three RDRs, previously known as ZmRDR3 and ZmRDR4 [32], to be ZmRDR6a, ZmRDR6b, and ZmRDR6c. ZmRDR6, such a multiple membered family, can be better revealed by identifying their double/triple mutants. Similar to how RDR6 was identified to be important in production of tasiRNAs, an unidentified RDR is expected to play a key role in production of phasiRNAs in maize [12].

## 3. Functions of miRNAs and phasiRNAs in Maize

### 3.1. The Interaction of miR156 and miR172 Fine Tunes Plant Developmental Timing

In maize, the transition from juvenile to adult leaves is marked by changes in cell shape, the production of epidermal wax deposits and of specialized cell types like leaf hairs, and a change in the identity of organs that grow from their axillary meristems. In maize and *Arabidopsis*, the roles of miR156 and miR172 interaction in developmental transitions have been widely explored [25,27,74,75,76]. MiR156 expression levels decrease with leaf age, while that of miR172 increase (Figure 5A). Their targets, encoding squamosa promoter binding protein-like (SBP-Like) and Apetala 2 (AP2) transcription factors, respectively, are expressed in complementary patterns. The mutant *Corngrass1* (*Cg1*) with increased levels of miR156 and reduced miR172 activity, displays restrained developmental transitions, prolonged juvenile features and delayed flowering (Figure 4) [25,77]. In turn, releasing *SPLs* from miR156 regulation leads to premature acquisition of adult leaf features and early flowering, resembling phenotypes of *glossy15* (*gl15*) plants, with reduced activity of miR172 targets (Figure 4) [27,29].

### 3.2. Plant Architecture Modulated by miR156 and miR319

In maize, plant architecture is mainly determined by tillers, plant height, leaf number, leaf angle and tassel branches. Compared with its ancestor, teosinte (*Zea mays* ssp. parviglumis), maize exhibits a profound increase in apical dominance with a single tiller [78]. Previous researches have proved *teosinte branched1* (*tb1*) gene, encoding a TCP transcription factor that is targeted by miR319, as a major contributor to this domestication change in maize (Figure 4 and Figure 5B) [79,80]. By increasing JA levels, the *tb1* mutant of maize causes a complete loss of apical dominance, allowing the unrestrained outgrowth of axillary buds and inflorescent architectural alterations [79,81]. MiR156 has been proved to be the important regulator in maize and rice plant architecture formation [25,82]. The dominant *Corngrass1* (*Cg1*) mutant of maize has phenotypic changes that are present in the grass-like ancestors of maize, exhibiting numerous tillers, inflorescent architectural alterations and erect leaves (Figure 5B) [25]. The research by Lu et al. [83] in rice revealed that the *ideal plant architecture1* (*IPA1*, *OsSPL14*) could directly bind to the promoter of rice *teosinte branched1* (*Ostb1*), to suppress rice tillering. Likewise, the maize tillering related *ZmSPL* (miR156 target) gene is possible at the upstream of *tb1* in related regulatory pathway. The roles of miR156 in leaf angle and inflorescent architectural modulation have been identified in the corresponding *ZmSPL* mutants, such as *LIGULELESS1* (*LG1*), *tasselsheath4* (*tsh4*, *ZmSBP2*), *UNBRANCHED 2* (*UB2*) and *UB3* [30,33,84,85]( Figure 4; Figure 5B).

### 3.3. Roles of miR172, miR156 and miR159 in Sex Determination

In maize, inflorescence development and sex determination are key factors for grain yield. MiR172 has been identified to play important roles in inflorescence development and sex determination (Figure 4) [86]. Especially, the interplay of miR156 and miR172 contributes largely in maize sex determination and meristem cell fate. In *Cg1* mutant, increased levels of miR156 cause similar phenotypic alterations as seen in *ts4* mutants [25]. Moreover, *STTMmiR172* and *ts4* mutants have reduced expression of miR172 and increased expression of at least two of its targets, *ids1* (*indeterminate spikelet1*) and *sid1* (*sister of indeterminate spikelet1*). These mutants displayed irregular branching within the inflorescence and feminization of the tassel caused by a lack of pistil abortion [86,87,88]. Decreased levels of miR156 have been detected in feminized tassels of maize *mop1* and *ts1* (*tasselseed1*), implying the missing link of miR156-SPLs with sex-determination genes *ts1*, *ts2*, *ts4*, *Ts6*, and *mop1* [34,89]. Additionally, the mutants of *fuzzy tassel* (encoding dicer-like1 protein) exhibit indeterminate meristems, fasciation, and alterations in sex determination [22]. Such reproductive development alterations are possibly associated with miR159-*GAMYB* pathway, with miR159 and its targets playing the important roles in another development [21,90].

### 3.4. Leaf Patterns Are Shaped by miR166 and miR390-TAS3

Leaves are the most important photosynthetic organs in land plants, which are nearly flat organs designed to efficiently capture light and perform photosynthesis. In maize the specification of abaxial/adaxial polarity was found to be intimately associated with sRNAs, such as miR166, miR390 and *TAS3* (Figure 4; Figure 5C) [26,38,91,92]. The miR166 targets belong to class III homeodomain/leucine zipper (*HD-ZIPIII*) genes. The maize miR166 knockdown and miR166 target over-expression mutants, *STTMmiR166* and *rolled leaf1* (*rld1*), displays an upward curling of the leaf blade that causes adaxialization or partial reversal of leaf polarity [26,88]. The roles of miR166 and *HD-ZIPIII* in leaf polarity are conserved between *Arabidopsis* and maize [93,94]. In plants, miR390 triggers *TAS3*-tasiRNA biogenesis, which interplay with ARF3 to take part in plant development regulation [54,67]. In maize, the mutants of tasiRNA biogenesis pathway components exhibit leaf polarity alterations, *ragged seedling2* (*rgd2*) or *leaf bladeless1* (*lbl1*) [13]. Moreover, several researches proposed that miR390-*TAS3* define the adaxial side of the leaf by restricting the expression domain of miR166, which in turn demarcates the abaxial side of leaves by restricting the expression of adaxial determinants [38,92,95].

### 3.5. PhasiRNAs and Maize Male Fertility

In hybrid maize, male sterility has been widely studied due to both its biological significance and commercial use in hybrid seed production [96]. Maize male fertility is determined by dozens of genes and sRNAs, especially phasiRNAs (Figure 2A,B and Figure 4) [11,12,96]. Indeed, a study reported that two classes of phasiRNAs, 21 and 24 nt in length, were detected to be highly expressed in maize anthers and confer male fertility [12]. The mutant lacking 21 nt phasiRNA, *ocl4*, showed male sterility due to defects in epidermal signaling. Meanwhile, the mutant lacking 24 nt phasiRNA lacking mutants also showed male sterility for due to defective anther subepidermis. This indicated that two types of phasiRNAs regulate anther development independently, with 21 nt premeiotic phasiRNAs regulating epidermal and 24 nt meiotic phasiRNAs regulating tapetal cell differentiation [12].

### 3.6. Other miRNA Functions in Maize

Several miRNAs have been identified to regulate important maize agronomic traits, such as kernel development, plant growth, abiotic stress tolerance, root development, and nutrition metabolism (Figure 4). The miR156 target, *tga1*, not only confers the domestication of maize naked grains, but also determines the maize kernel shape and size [97,98]. A report on *ZmGRF10*, a miR396 target, indicated that this miRNA is a potential regulator for maize leaf size and plant height [99]. The overexpression *ZmGRF10* mutant displayed reduction in leaf size and plant height by decreasing cell proliferation. Other studies have shown that drought and salinity stresses induce aberrant expression of many miRNAs in maize, for example miR166 and miR169. In maize, miR169 plays a critical role during plant drought, salt and ABA stress response by targeting *NUCLEAR FACTOR-Y subunit A* (*NF-YA*) genes [28]. In *Arabidopsis* and rice, miR166-*HD-ZIP IIIs* have been proven to be associated with drought and ABA stress resistance through maintaining ABA homeostasis [100,101]. Based on our unpublished data, the maize miR166 probably affects tolerance to drought and salinity stresses like in rice and *Arabidopsis*. In maize, miR164 was experimentally identified to be an important regulator in lateral root development by targeting *ZmNAC1* [31,102].

A recent research identified miR528, a monocot-specific miRNA, to be an important regulator for maize nitrogen metabolism in maize. In the miR528 knock-down mutant of maize, under nitrogen-luxury conditions, targets of miR528 are upregulated and mediate increase in lignin content along with superior lodging resistance [20]. The miR399 was identified to regulate the low-phosphate responses in maize [103]. The transgenic plant with miR399 over-expression showed significant phosphorus-toxicity phenotypes, indicating that miR399 is functionally conserved in monocots and dicots.

## 4. Exploiting the Roles of Maize Small RNAs in Important Agronomic Traits Improvement

Most of agronomic traits are quantitative traits, which are controlled by multiple loci and complex regulatory networks. MiRNAs and phased siRNAs are important participants in these complex regulatory networks. Manipulating the expression levels of miRNAs, miRNA targets, and phased siRNAs is a possible way for agronomic traits improvement. With grain yield increasing, the agronomic traits of maize have been improved through genetic selection [104], which is probably consistent with the elite allele selection of miRNAs and their targets in breeding. Compared with old maize varieties, modern varieties usually have reduced stature, more upright leaves, decreased tassel size, rolling leaf, superior staygreen, less tillers, shorter anthesis-silking interval, less ears per plant and superior stress resistance [104]. Based upon the knowledge about miRNA and phased siRNA functions, manipulating the expression of these small regulatory RNAs and their targets is a possible approach for agronomic traits improvement.

Flowering time represents the developmental transition from vegetative to reproductive phase. Maize spread from its origin to worldwide places with the gradually adaption of flowering time to the local climate [105]. Flowering time determines the length of vegetative phase, biomass and grain yield in maize. The interplay between miR156 and miR172 fine tunes the maize developmental timing and tillering [25,86,88]. Increasing the expression levels of miR156 can elongate the vegetative phase and tillering in maize, which is important to achieve high biomass for silage feed. MiR156 silencing or miR172 over-expression is able to impel maize flowering and precocity, which is in favor of maize mechanized harvest in special regions. 

Ideal plant architecture is highly associated with maize planting density and lodging resistance, thereby achieving higher yield. Maize miR156 also regulates plant architectural traits through binding its target genes, such as *tsh4*, *LG1*, *UB2* and *UB3* [30,33,84,85,106,107]. Manipulating the expression of these *ZmSBPs* at optimal levels is needed for idea plant architectural traits. For instance, decreased expression of *LG1* can promote the leaf angle and reduce tassel branches. Furthermore, manipulating the expression of *UB2* and *UB3* in tassel branches and ear rows by using tissue-specific promotor is helpful to get ideal tassel and ear architecture. Additionally, repressing the expression of miR166 or increasing the expression of its targets will increase the leaf rolling, which can be helpful for improving the leaf shapes [88].

In global maize production, lodging and drought are two main abiotic stresses that accounts for large yield loss annually. In a recent research, miR528 has been proved to affect lodging resistance through regulating lignin biosynthesis [20]. Gene silencing of miR528 or overexpression of its targets is helpful for enhancing maize lodging resistance. Knock-down of miR164 promotes maize lateral root development, which can help toward drought and lodging resistance [31]. In the response toward abiotic stress, such as drought, ABA and salinity, miR169 and its targets (NF-YAs) contribute the major regulatory roles through ABA signaling [108]. Lowering the expression of miR169, or increasing that of NF-YAs, can facilitate maize resistance to drought. MiR166 silencing confers resistance against drought in rice and *Arabidopsis*, which is likely conserved in maize too [88,100,101].

## 5. Future Perspectives

As discussed above, enhanced knowledge on miRNA and phased siRNA functions will be helpful for improving some agronomic traits, including developmental timing, plant architecture, and abiotic stress resistance. Genetic engineering for elite maize germplasms and hybrids still face several hurdles. First, only a small proportion of miRNAs and phased siRNAs have been studied in maize, their complex regulatory networks remain largely unknown. The functional identification of sRNAs is largely dependent upon creating mutants. In maize, the abundant genetic variations or mutations in germplasm pools can provide useful raw materials for the study of these regulatory sRNAs or their targets [109]. Creating new mutants for specific sRNA using artificial miRNA, Short tandem target mimic (STTM) or target mimic (TM) techniques, are efficient strategies for uncovering the functions of these regulatory sRNAs in maize [110,111,112]. Second, plant miRNA and phased siRNA usually express in spatial and temporal manner. Thus, manipulating the expression of miRNA and phased siRNA in specific tissues and developmental stages can precisely target the traits for improvement. This can be achieved by expressing the transgene expression using tissue- or development-specific promoters, or inducible promoters. Fine genome editing of miRNAs, phased siRNAs and target genes by the CRISPR/Cas9 system can facilitate more subtle manipulations for the target agronomic traits, which is an alternative strategy. Third, for maize hybrids worldwide planted, screening elite hybrid is the most important mask in maize breeding. Usually, the ideal phenotypes in parental inbred lines do not always transfer to the corresponding hybrid. Screening of an elite hybrid is bit of an art and magic, which requires all the yield related traits to reach a balance, and with high heterosis and stress resistance. The current theory of heterosis model facilitate the breeders to make hybrid crosses with high heterosis. Screening the inbred lines with elite genotype/haplotype of miRNAs, phased siRNAs and their targets is fundamental in breeding. Introgressing the elite genotype or haplotype into inbred lines based on heterosis model/heterotic groups will enable the parental elite phenotypes get transferred to their hybrids. 

## Figures and Tables

**Figure 1 plants-08-00170-f001:**
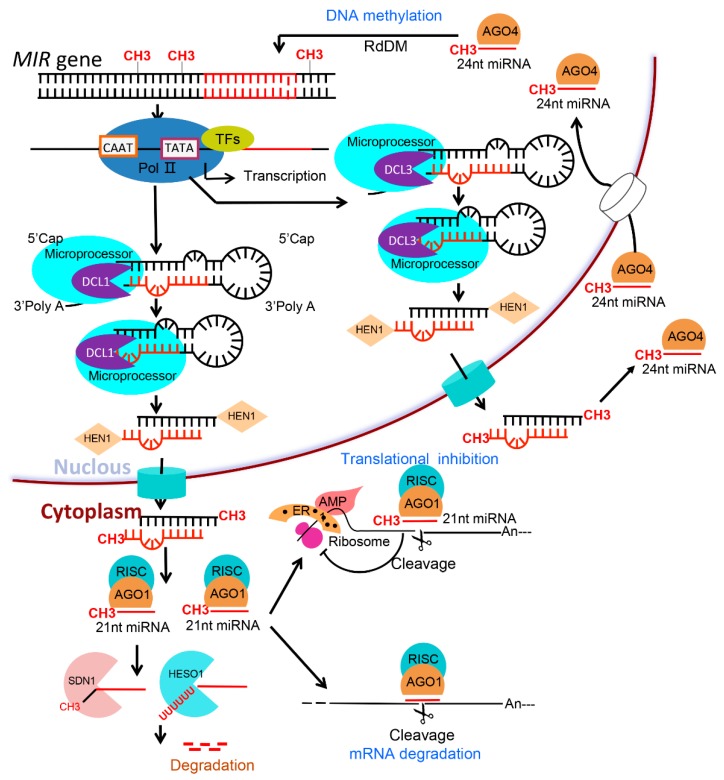
An overview showing micro RNA (miRNA) biogenesis and functioning in plants. *MIRNA* genes are transcribed to form primary miRNAs, from which 21 and 24 nt miRNAs are processed by DCL1 and DCL3, respectively. Their 3′ ends are methylated by HEN1. While the 21 nt species are involved in cleavage or translational inhibition of the target mRNAs, the 24 nt miRNAs are involved in DNA methylation.

**Figure 2 plants-08-00170-f002:**
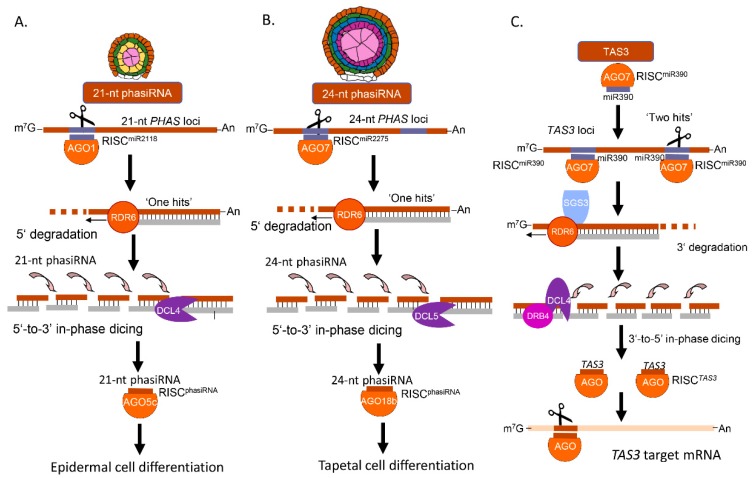
Phased small interfering RNA (phasiRNA) and trans-acting simple interfering RNA (tasiRNA) biogenesis pathways in maize. (**A**) The 21 nt phasiRNA biogenesis pathway. (**B**) The 24 nt phasiRNA biogenesis pathway. The regulatory mechanism of 24 nt has not been fully uncovered. (**C**) *TAS3*-tasiRNA biogenesis pathway.

**Figure 3 plants-08-00170-f003:**
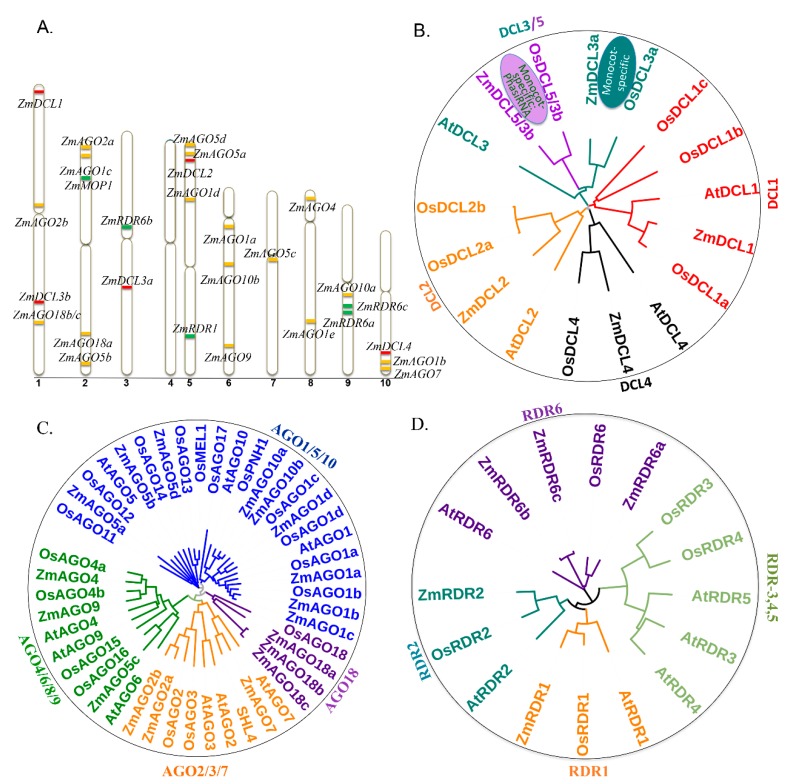
Chromosomal locations and phylogenetic analysis of known and putative components of the RNAi and miRNA pathways in maize. The protein sequences of dicer-like (DCL), argonautes (AGOs) and RNA-dependent RNA polymerase (RDR) in *Arabidopsis*, rice and maize AGOs were obtained from protein database (http://www.ncbi.nlm.nih.gov/protein). The neighbor-joining tree was constructed using Clustal omega [70] and iTol online software [71]. (**A**) The abbreviation of *At* represents *Arabidopsis thaliana*, *Os* for *Oryza sativa*, and *Zm* for *Zea mays*. Red bars indicate the chromosomal locations of *ZmDCLs*, yellow bars for *ZmAGOs*, and green bars for *ZmRDRs*. (**B**) Plants have four types of DCL proteins. There are 4 DCLs encoded in *Arabidopsis* genome, 4 DCL family members in rice, and 4 DCLs in maize. Of these DCL proteins, DCL3a and DCL3b are considered specific to monocots and predate the divergence of rice and maize. (**C**) 10, 19 and 18 AGOs are encoded by *Arabidopsis*, rice and maize, respectively, that can be divided phylogenetically into five subgroups in maize: AGO1, MEL1/AGO5, AGO7, AGO4, and AGO18. AGO18 subgroup has three members in maize. ZmAGO18a-c are considered specific to monocots along with OsAGO18. (**D**) Plants have six types of functionally distinct RDRs. While *Arabidopsis* has all the six types, rice lacks RDR5 and maize lacks RDR3, 4, and 5. In contrast to *Arabidopsis* and rice which have only single member RDR6 family, maize has a multiple member RDR6 family, which is composed of ZmRDR6a, ZmRDR6b, and ZmRDR6c.

**Figure 4 plants-08-00170-f004:**
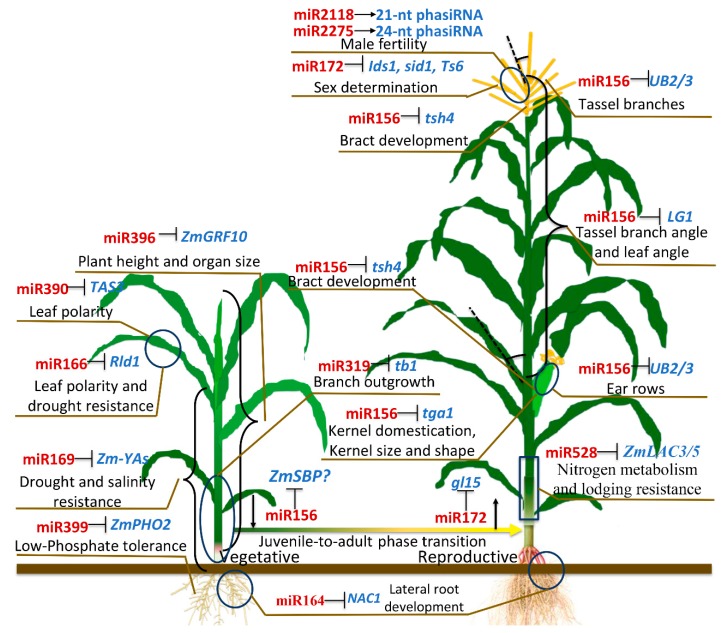
Summary of functionally validated miRNAs and their targets in maize. Nine miRNAs (in red font) and three phasiRNAs (in red font) that regulate specific agronomic traits (in black font) by inducing their targets (in blue font) gene silencing. Question marks indicate that specific *ZmSBP* taking part in juvenile-to-adult phase transition is not known.

**Figure 5 plants-08-00170-f005:**
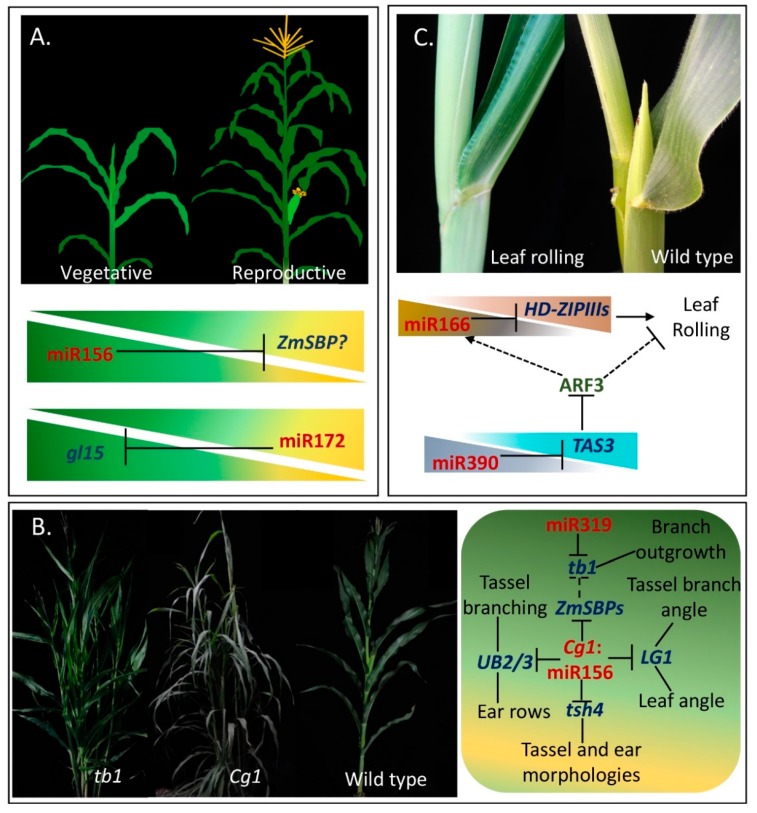
miRNAs or miRNA-phasiRNA interactions in agronomic traits. (**A**) Maize plant developmental timing is fine-tuned by the interaction between miR156 and miR172. However, the misslink between the two miRNAs still need to be addressed; (**B**) plant architectural modulation by the interaction of miR156 and miR319. Representative plants, *tb1* (leaf), *Cg1* (**middle**), and wild type (**right**) (all in the background of Chinese inbred line Zheng58) (Unpublished data), are shown. The potential *ZmSBP* gene probably connects the phenotype of apical dominance loss between maize mutants *tb1* and *Cg1.* In this context, the connection between *ZmSBPs* and *tb1* still need to be experimentally identified; (**C**) leaf shapes are being regulated by miR166 and miR390-TAS3 regulatory networks. *STTMmiR166* mutants have rolling leaf phenotype (**left**), the wild type is ZZC01 (**right**). In this context, the connection between miR166 and ARF3 is still unclear.

**Table 1 plants-08-00170-t001:** Known and putative components of the micro RNA (miRNA) and simple interfering (siRNA) pathways in maize.

Gene	Accession Number	Chromosomal Location (5′-3′)	Type
1. *ZmDCLs*			
*ZmDCL1*	GRMZM2G040762_P01	Chr. 1: 4,600,841–4,608,248	DCL1
*ZmDCL2*	GRMZM2G301405_P01	Chr. 5: 19,916,753–19,927,967	DCL2
*ZmDCL3a*	GRMZM5G814985_P01	Chr. 3: 164,415,209–164,418,189	DCL3
*ZmDCL5/3b*	GRMZM2G413853_P01	Chr. 1: 229,801,762–229,819,069	DCL3
*ZmDCL4*	GRMZM2G160473_P01	Chr. 10: 129,990,456–129,992,917	DCL4
2. *ZmAGOs*			
*ZmAGO1a*	GRMZM2G441583_P01	Chr. 6: 43,253,105–43,261,555	AGO1
*ZmAGO1b*	AC209206.3_FGP011	Chr. 10: 137,506,877–137,513,415	AGO1
*ZmAGO1c*	GRMZM2G039455_P01	Chr. 2: 17,563,301–17,573,156	AGO1
*ZmAGO1d*	GRMZM2G361518_P01	Chr. 5: 64,791,077–64,796,881	AGO1
*ZmAGO2a*	GRMZM2G007791_P01	Chr. 2: 9,973,816–9,981,340	ZIPPY
*ZmAGO2b*	GRMZM2G354867_P01	Chr. 1:142,397,812–142,403,450	ZIPPY
*ZmAGO4*	GRMZM2G589579_P01	Chr. 8: 2,511,663–2,519,008	AGO4
*ZmAGO5a*	GRMZM2G461936_P02	Chr. 5: 13,611,800–13,618,698	MEL1
*ZmAGO5b*	GRMZM2G059033_P01	Chr. 2: 233,385,077–233,392,000	MEL1
*ZmAGO5c*	GRMZM2G347402_P01	Chr. 7: 72,044,775–72,053,779	MEL1
*ZmAGO5d*	GRMZM2G123063_P01	Chr. 5:4,000,995–4,009,425	MEL1
*ZmAGO7*	GRMZM2G354867_P01	Chr. 10: 141,823,070–141,828,449	ZIPPY
*ZmAGO9*	GRMZM2G141818_P03	Chr. 6: 168,642,369–168,650,358	AGO4
*ZmAGO10a*	AC189879.3_FG003	Chr. 9: 87,408,375–87,414,276	AGO1
*ZmAGO10b*	GRMZM2G079080_P02	Chr. 6: 103,286,236–103,293,200	AGO1
*ZmAGO18a*	GRMZM2G105250_P01	Chr. 2: 199,510,528–199,516,085	OsAGO18
*ZmAGO18b*	GRMZM2G457370_P01	Chr. 1: 250,132,189–250,137,737	OsAGO18
*ZmAGO18c*	GRMZM2G457370_P02	Chr. 1: 250,132,189–250,137,737	OsAGO18
3. *ZmRDRs*			
*ZmRDR1*	GRMZM2G481730_P01	Chr. 5: 205,385,818–205,389,710	RDR1
*ZmMOP1*	GRMZM2G042443_P01	Chr. 2: 41,131,324–41,136,928	RDR2
*ZmRDR6a*	GRMZM2G357825_P01	Chr. 9: 109,055,576–109,093,885	RDR6
*ZmRDR6b*	GRMZM2G145201_P01	Chr. 3: 102,532,883–102,536,036	RDR6
*ZmRDR6c*	GRMZM2G347931_P01	Chr. 9: 106,302,354–106,306,175	RDR6

Note: this information for maize dicer-like (DCL), argonaute (AGO), and RNA-dependent RNA polymerase (RDR), including accession number, chromosomal location and ORFs, was retrieved from the B73 maize sequence database (http://www.maizesequence.org/index.html).

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
