# Peer review of "Perspectives on microRNAs and Phased Small Interfering RNAs in Maize (Zea mays L.): Functions and Big Impact on Agronomic Traits Enhancement"

_plants, 2019, doi:10.3390/plants8060170_

Round 1
Reviewer 1 Report
This review updates on the current status of maize miRNAs and PhasiRNA and discusses on their potential application in agronomic trait improvement in maize. The manuscript provides a lot of information and can be accepted for publication. My major concern is on the English language. It cannot be accepted for publication in the current form and needs to be thoroughly reviewed by a professional English editor or a native English speaking scientist.
Detail comments
-Page 3, para 2, lines 17 to 24-citation is missing or not correctly cited
-Figure 5b. where is the source of this figure-citation is missing?
-Abstract Line 8 functional'? Functionally?
-Abstract Line 12, 'maize has limited studies'-the language needs to be corrected
-Introduction page 1 Para 1, Line 1, 'In plants, sMRA are short---' are plant sRNA different from animal sRNAnin size?
-Page 3, Line 3, 'This indicts? Indicates?
-page 3, Line 8, 'PhasiRNAs represent a kind of', type of?
-page3, Line 9, 'that directed by', a verb is missing, also Page 4, L14
Page3, Line 15, 'and mediating transcriptional' mediate? also page 4, L15
Page3, Line 21, 'experimental' experimentally?
Page 4 Line 17, 'RDRs in rice....(17)? Are reported in rice?
Page 6, Line 6, needs rephrasing.
Page 6, Line 11, 'ZmDCL1 was identified...'showed high similarity with...?
Page 6, Line 20, 'one of most..' One of the most?
Page 6, Line 28, AtAGO7 preferentially associates...'is preferentially associated..'
Page 7, Line 11, GRMZM2G457370 encodes ZmARGO18b and ZmARGO18C? Are these genes different sequences? I looks like GRMZM2G45370 encode two genes?
Page 7, the sentence from Line 15 to Line 18 is vague. Divide into two?
Page 7, Line 22-23, 'Concurrence spatially and temporally'-spatial and temporal expression?
Page 8, Line 1, 'six, five and five....'needs rewriting
Page 9, Line 10, 'siRNAs that triggered by'...siRNA that are?
Page 9, Line 16, 'In grass, includes maize,'..In grasses, including?
Page 9, Line 21, 'are phased by...' regulated by?
Page 10, Line 5-6, the sentence stating with 'The mutants...' it is not clear. The two mutants, both, resulted in through...
Page 11, Line1-'Idea plant...'-Ideal?
Page 11 Line 17-18, The sentence starting with 'Likewise,....' is speculation.
Page 12, Line 2 'MiR172 and MiR159...' Is reference 89 correct?
Page 13, Line 3, 'the increasing researches....?Many researchers/reports? Same problem on Page 14 Line 5
Page 13, Linee 12, 'in length of 21 and 24, ...detected express', 21 and 24 in length ..shown to express?
Page 13, Line 14, 'Alternatively...' looks out of place
Page 13, Line 20, 'besides metioned above...'besides what has been mentioned?
Page 13, Line 28, 'A research for GRF10...' a report on?
Page 14 Line 1, 'Transgenic mutant...' mutant?
Page 14, Line 17-'signaling and the changing conditions...' Needs rewriting.
Page 15, Line 23,'there by to associate with...', affecting the?
Page 17, Line 9, 'In maize, the abundant, various...' it is vague
Page 17, Line 22, 'maize is a hybrid crop...'All maize?
Figure 34, Figure 1-Fonts are hard too small to read
Page 38. the figure legend is not easy to understand the figure. different colors need to be applied for different genes...
Author Response
Dear Editor and the Reviewer,
Thank you very much for your critical comments for our manuscript. We have revised the manuscript according to the reviewers’ comments. Moreover, the English language of this manuscript has been edited in detail by Dr. Sachin Teotia (a co-author of this manuscript).
Detailed responses are as following (in blue):
-Page 3, para 2, lines 17 to 24-citation is missing or not correctly cited
Thank you for pointing this out, citation was missing. We have cited the related references in the manuscript.
-Figure 5b. where is the source of this figure-citation is missing?
We have taken photos of maize tb1 and Cg1 plants (with the background of Chinese inbred line Zheng58) in the field. And we made this figure using these photos. Therefore, Figure 5b is an original figure.
-Abstract Line 8 functional'? Functionally?
We have revised this mistake in the manuscript
-Abstract Line 12, 'maize has limited studies'-the language needs to be corrected
We have rewritten this sentence in the manuscript and rewritten as, “In contrast to Arabidopsis and rice, studies on biogenesis and functions of miRNAs and phasiRNAs in maize are limited, which restricts the small RNA-based fundamental and applied studies in maize”
-Introduction page 1 Para 1, Line 1, 'In plants, sMRA are short---' are plant sRNA different from animal sRNAnin size?
We have revised this improper expression.
-Page 3, Line 3, 'This indicts? Indicates?
We have revised this mistake in the manuscript.
-page 3, Line 8, 'PhasiRNAs represent a kind of', type of?
We have revised this improper usage in the manuscript.
-page3, Line 9, 'that directed by', a verb is missing, also Page 4, L14
We have revised these kind of grammatical mistakes in the manuscript.
Page3, Line 15, 'and mediating transcriptional' mediate? also page 4, L15
We have revised these two mistakes in the manuscript.
Page3, Line 21, 'experimental' experimentally?
We have revised the mistake in the manuscript.
Page 4 Line 17, 'RDRs in rice....(17)? Are reported in rice?
We have revised this improper description in the manuscript.
Page 6, Line 6, needs rephrasing.
We have deleted this sentence from the manuscript.
Page 6, Line 11, 'ZmDCL1 was identified...'showed high similarity with...?
We have addressed this point in the manuscript and now written as, “ZmDCL1 showed high similarity with Arabidopsis AtDCL1 and rice OsDCL1a–1c”..
Page 6, Line 20, 'one of most..' One of the most?
We have revised this mistake in the manuscript.
Page 6, Line 28, AtAGO7 preferentially associates...'is preferentially associated.'
We have revised this mistake in the manuscript as suggested.
Page 7, Line 11, GRMZM2G457370 encodes ZmARGO18b and ZmARGO18C? Are these genes different sequences? I looks like GRMZM2G45370 encode two genes?
GRMZM2G45370 have two transcripts that encode two proteins, ZmARGO18b and ZmARGO18c. We have added a description in the manuscript as, “The two transcripts of GRMZM2G105250 encodes ZmAGO18a and ZmAGO18b,, and GRMZM2G457370 encodes ZmAGO18c
Page 7, the sentence from Line 15 to Line 18 is vague. Divide into two?
We have rewritten this sentence in the manuscript.
Page 7, Line 22-23, 'Concurrence spatially and temporally'-spatial and temporal expression?
We have revised this mistake in the manuscript as suggested
Page 8, Line 1, 'six, five and five....'needs rewriting
We have revised this mistake in the manuscript and written as, “Corresponding to six and five RDRs in Arabidopsis and rice, respectively, five have been identified in maize”.
Page 9, Line 10, 'siRNAs that triggered by'...siRNA that are?
We have removed this line from the manuscript.
Page 9, Line 16, 'In grass, includes maize,'.In grasses, including?
We have revised this grammatical mistake in the manuscript.
Page 9, Line 21, 'are phased by...' regulated by?
We have revised this misuse in the manuscript and written as, “DCLs further process dsRNAs to produce 21 or 24 nt phasiRNAs.”
Page 10, Line 5-6, the sentence stating with 'The mutants...' it is not clear. The two mutants, both, resulted in through...
We have rewritten this sentence according to reviewer’s advice.
Page 11, Line1-'Idea plant...'-Ideal?
We have revised this misuse in the manuscript.
Page 11 Line 17-18, The sentence starting with 'Likewise,....' is speculation.
We have revised this sentence in the manuscript and written as” Likewise, the maize tillering related ZmSPL (miR156 target) gene acts upstream of tb1.”
Page 12, Line 2 'MiR172 and MiR159...' Is reference 89 correct?
We have checked carefully. Only miR172 was experimentally identified to play important roles in inflorescence development and sex determination. We have revised this improper description and removed miR159 from the statement.
Page 13, Line 3, 'the increasing researches....? Many researchers/reports? Same problem on Page 14 Line 5
We have revised the two points in the manuscript as suggested.
Page 13, Line 12, 'in length of 21 and 24, ...detected express', 21 and 24 in length ..shown to express?
We have revised this logical error in the manuscript.
Page 13, Line 14, 'Alternatively...' looks out of place
We have used ‘furthermore’ to replace it in the manuscript.
Page 13, Line 20, 'besides mentioned above...'besides what has been mentioned?
We have deleted the improper expression from the manuscript.
Page 13, Line 28, 'A research for GRF10...' a report on?
We have revised this point according the reviewer’s advice.
Page 14 Line 1, 'Transgenic mutant...' mutant?
We have rewritten this sentence in the manuscript as just “mutant”.
Page 14, Line 17-'signaling and the changing conditions...' Needs rewriting.
We have rewritten this sentence in the manuscript.
Page 15, Line 23,'there by to associate with...', affecting the?
We have revised this point according the reviewer’s advice and written as, “Flowering time determines the length of vegetative phase, biomass and grain yield in maize.”
Page 17, Line 9, 'In maize, the abundant, various...' it is vague
We have rewritten this sentence in the manuscript.
Page 17, Line 22, 'maize is a hybrid crop...'All maize?
We have revised this improper expression in the manuscript.
Page 34, Figure 1-Fonts are hard too small to read
We have re-drawn Figure 1 to make it easy to follow.
Page 38. The figure legend is not easy to understand the figure. Different colors need to be applied for different genes...
We have revised the figure legends to make it easy to understand. And we used different colors to represent different genes.
Reviewer 2 Report
The review lacks a clear classification of different sRNA classes that would guide the reader though the whole manuscript. Indeed the review is very confused, it mixes up different concepts and information without any clear definition both of pathways and genes involved in sRNA production and in regulation of their functions. Please follow a classification from the literature and then always refer to it in the whole manuscript. In my opinion, the authors before going into details of different enzymes participating in sRNA biogenesis, should summarize these pathways and then highlight differences between maize and other model species.
In addition the manuscript is very long and should be shortened. Indeed many concepts (see for instance the definition of phasiRNAs) are repeated many times in the review. Of course there are many data on miRNAs and their targets, but only the main work/results should be reported and emphasized.
Please check the english language.
Author Response
Dear Editor and the Reviewer,
Thank you very much for your critical comments for our manuscript. We have revised the manuscript according to the reviewers’ comments. Moreover, the English language of this manuscript has been edited in detail by Dr. Sachin Teotia (a co-author of this manuscript).
Detailed responses are as following (in blue):
1. Please follow a classification from the literature and then always refer to it in the whole manuscript.
We have checked the related references carefully. And, followed a classification of small RNAs that was proposed by Axtell, M.J. (2013). Then, this literature is referred in the whole manuscript.
2. In my opinion, the authors before going into details of different enzymes participating in sRNA biogenesis, should summarize these pathways and then highlight differences between maize and other model species.
According to the reviewer’s advice, we have arranged the context of the manuscript. First, we listed the core components of sRNA biogenesis in plants. Then, we summarized the sRNA biogenesis pathways in maize, and highlighted the differences between maize and other model species. Third, we analyzed three key participants in sRNA biogenesis.
3. The manuscript is very long and should be shortened.
We read the manuscript in detail, and some meaningless descriptions, repeated concepts and unrelated ideas are deleted. Additionally, we have only reviewed the main works/results in context of functions of maize miRNAs and phasiRNAs.
4. Indeed many concepts (see for instance the definition of phasiRNAs) are repeated many times in the review.
We have checked this point in whole manuscript, and deleted the repetitions.
5. There are many data on miRNAs and their targets, but only the main work/results should be reported and emphasized.
We have revised the manuscript according to the reviewer’s advice.
6. Please check the English language.
The co-author Sachin Teotia has revised this manuscript in detail.
Round 2
Reviewer 2 Report
This new version of the review has been improved. However I still have a few concerns.
In describing the pathways producing sRNAs in in model plants and maize, the RdDM pathway has been completely omitted. I understand the authors want to focus on miRNAs and phased RNAs, however RdDM has been investigated in maize and is cited in the review by the authors but not described at all.
The review still contains many examples of miRNAs, phased RNAs and references regarding Arabidopsis and rice, which could be omitted, to better focus the attention on what is known about maize.
Some minor points:
page 4 line 20 "Among them, only a few of them" sounds redundant.
page 5 line 17 twice Next
page 9 line 22 please check the sentence
There are some words in red color throughout the manuscript: what's the mining?
Author Response
Dear the Reviewer,
Thanks very much for your critical comments for our manuscript. We have revised the manuscript according to the comments, and tracked the revision changes.
Detail revisions are as following,
1. In describing the pathways producing sRNAs in in model plants and maize, the RdDM pathway has been completely omitted. I understand the authors want to focus on miRNAs and phased RNAs, however RdDM has been investigated in maize and is cited in the review by the authors but not described at all.
According to the reviewer’s advice, we have added RdDM pathway in Fig. 1 and had a revision in manuscript.
2. The review still contains many examples of miRNAs, phased RNAs and references regarding Arabidopsis and rice, which could be omitted, to better focus the attention on what is known about maize.
We have checked the manuscript in detail, and deleted descriptions and references regarding Arabidopsis and rice as possible.
3. Page 4 line 20 "Among them, only a few of them" sounds redundant.
We have revised this mistake.
4. Page 5 line 17 twice Next
We have revised this type error.
5. Page 9 line 22 please check the sentence
We have re-written this sentence.
6. There are some words in red color throughout the manuscript: what's the meaning?
These words in red color are revised in the former revision, but we forgot to change the font color. We have revised this kind of mistake in the manuscript.